# Self-Supported Biopolymeric Films Based on Onion Bulb (*Allium cepa* L.): Gamma-Radiation Effects in Sterilizing Doses

**DOI:** 10.3390/polym15040914

**Published:** 2023-02-11

**Authors:** Marco Antonio da Costa Borges, Amanda Rinaldi Sorigotti, Rafaella Takehara Paschoalin, José Alberto Paris Júnior, Lucas Henrique Domingos da Silva, Diógenes Santos Dias, Clóvis Augusto Ribeiro, Elmo Silvano de Araújo, Flávia Aparecida Resende, Hernane da Silva Barud

**Affiliations:** 1Laboratory of Biopolymers and Biomaterials (BIOPOLMAT), University of Araraquara (UNIARA), Araraquara, São Paulo 14801-340, Brazil; 2Graduate Program in Materials Science and Engineering (PPGCEM), Federal University of São Carlos (UFSCAR), São Carlos, São Paulo 13565-905, Brazil; 3Biosmart Nanotechnology, Araraquara, São Paulo 14808-162, Brazil; 4Chemistry Institute (IQ), São Paulo State University (UNESP), Araraquara, São Paulo 14800-060, Brazil; 5Department of Nuclear Energy (DEN), Federal University of Pernambuco (UFPE), Recife, Pernambuco 50670-901, Brazil

**Keywords:** onion-based films, casting, effect of washing, sterilization, γ-radiation

## Abstract

Sterilization is a fundamental step to eliminate microorganisms prior to the application of products, especially in the food and medical industries. γ-irradiation is one of the most recommended and effective methods used for sterilization, but its effect on the properties and performance of bio-based polymers is negligible. This work is aimed at evaluating the influence of γ-radiation at doses of 5, 10, 15, 25, 30, and 40 kGy on the morphology, properties, and performance of bioplastic produced from onion bulb (*Allium cepa* L.), using two hydrothermal synthesis procedures. These procedures differ in whether the product is washed or not after bioplastic synthesis, and are referred to as the unwashed hydrothermally treated pulp (HTP) and washed hydrothermally treated pulp (W-HTP). The morphological analysis indicated that the film surfaces became progressively rougher and more irregular for doses above 25 kGy, which increases their hydrophobicity, especially for the W-HTP samples. In addition, the FTIR and XRD results indicated that irradiation changed the structural and chemical groups of the samples. There was an increase in the crystallinity index and a predominance of the interaction of radiation with the hydroxyl groups—more susceptible to the oxidative effect—besides the cleavage of chemical bonds depending on the γ-radiation dose. The presence of soluble carbohydrates influenced the mechanical behavior of the samples, in which HTP is more ductile than W-HTP, but γ-radiation did not cause a change in mechanical properties proportionally to the dose. For W-HTP, films there was no mutagenicity or cytotoxicity—even after γ-irradiation at higher doses. In conclusion, the properties of onion-based films varied significantly with the γ-radiation dose. The films were also affected differently by radiation, depending on their chemical composition and the change induced by washing, which influences their use in food packaging or biomedical devices.

## 1. Introduction

The onion (*Allium cepa* L.) is a widely cultivated and consumed vegetable. Physiologically, the onion is an underground modification of the stem, a bulb represented by adventitious roots on the underside and fleshy (edible) leaves arranged concentrically on the upper side of the reduced stem. A sheath of dry membranous scales covers the bulb, which is called a tunic (colored, dry outer protective skin) [1,2]. The onion cell wall is composed of structural carbohydrates (a), primary cellulose consisting of microfibrils, (b) xylans, hemicellulose embedded in the pectin matrix, and (c) pectic polysaccharide, consisting mainly of Rhamnogalacturas II or RG-II. Hydrothermal treatment transforms the structural carbohydrates of onion biomass into bioplastic, with pectin acting as a natural binding agent [3].

Raw onion consists of 90% water, 1.5% soluble proteins, 0.1% fat, 8.7% structural and non-structural carbohydrates [4,5,6], flavonoids in low concentration [7], anthocyanins [8], tearing and tasty compounds such as sulfonic acids, ammonia [9,10,11], secondary metabolites such as S-methyl-cysteine sulfoxide (MCSO, Methylin, S-allyl-cysteine sulfoxide, S-transprop-1-enyl-cysteine sulfoxide) and S-propylcysteine sulfoxide [9,12], and presence of high levels of galacturonic acids [13,14].

Onion-based bioplastics can be produced using hydrothermal processing methods and, due to a favorable interaction with cells, have been studied as a potential support for tissue engineering and regenerative medicine. This is the case with some formulations for topical treatment of healing [15], food packaging [16], as well as a new biopolymer obtained and used to produce sustainable substrates for flexible organic light-emitting diodes FOLEDs [17]. Barreto [18] evaluated the cytotoxic and mutagenic profiles of *Allium cepa* L. polymer films obtained by the casting process, proving their safety for primary food packaging [18]. Soares [19] investigated the effect of edible onion (*Allium cepa* L.) films on beef burgers’ quality, sensory properties, and shelf life.

To expand the application of onion-based bioplastics in the food or medical industries, they must be sterilized by a method that maintains their physical and biological properties. Gamma radiation is an up-and-coming and efficient sterilization agent for several pharmaceuticals, cosmetics, biomaterials, and polymers that are sensitive to heat and incompatible with certain sterilizing substances [20].

Onion film sterilization techniques have not been extensively studied, and research on radiation sterilization remains limited. However, radiation has become an alternative commercial sterilization technique to eliminate or reduce pathogenic organisms and improve product quality and shelf life [21]. Furthermore, it has potential benefits over traditional sterilization techniques, such as chemicals and heat, due to its deep penetration into the material [21,22].

There are a variety of effective sterilization techniques to promote the elimination of microorganisms. Moist and dry heat, filtration, gas plasma, ethylene oxide, and radiation are techniques that can be cited as sterilants [23,24]. The method used for sterilization must be compatible with the item to be sterilized to avoid damage, mainly to the physicochemical properties of the product [25]. Among these sterilization methods, gamma radiation can be applied in industry, both in polymeric, pharmaceutical, food, and medical and surgical materials [26]. It is less usual in the case of non-ionizing radiation (ultraviolet rays), because it has a low penetrating power and does not pass through glass, dirty films, or other thick materials. Ethylene oxide (ETO) sterilization is a colorless, volatile gas that is effective for sterilizing hospital medical supplies because it is highly penetrating yet highly toxic and carcinogenic [27,28]. Thus, gamma radiation has excellent advantages as a sterilization method, as it can penetrate more inaccessible places. It also has negligible heat production without the presence of residues, dispensing with quarantine, as well as excellent reliability [29], being economically appropriate and more efficient than ethylene oxide [30].

Maximum radiation doses, essential to achieve sterility, are defined for various food categories by FDA regulation under 21 CFR179.26 [31,32] which establishes the standard radiation dose of 25 kGy to sterilize products [21,33]; however, it is essential to investigate the effects of radiation on the physical properties of the bioplastic at doses close to the sterilization dose to obtain information on resistance to radiation.

Biopolymeric materials, submitted to sterilization by radiation, can suffer damages in their properties induced by the sterilization process [34]. Thus, post-radiation effects on materials are also necessary to characterize. This study focuses on the impact of Co-60 γ-radiation on the physical, chemical, thermal, and mechanical properties and mutagenicity of onion films with different dose levels of ionizing radiation intensity. In addition, the effects of radiation on biomaterials were compared with one another, with non-irradiated materials as the control.

## 2. Materials and Methods

### 2.1. Onion-Based Films Production

The production of the onion-based films was completed according to the methodology described previously by Dias et al. [3]. In summary, medium-sized yellow onion (*Allium cepa* L.) bulbs were commercially purchased and washed with tap water in order to remove the impurities derived from the harvest and transportation. After this initial procedure, the skin layers were removed, and then the bulbs were cut into four slices along the length, washed again, and hydrothermally treated. Before drying, the slices were inserted in an electric pressure cooker (PE-38, Mondial, Brazil) for 30 min at a temperature and pressure of 121 °C and 1.2 kgf cm^−2^, respectively. After that, the remaining liquid fraction was removed. This work used two distinct hydrothermal treatments to treat the pulps before film production. In the first treatment, named unwashed hydrothermally treated pulp (HTP), the autoclaved onion pulp was ground in an Ultra Turrax (Campinas, São Paulo, Brazil) disperser for 10 min at a frequency of 7000 rpm. In the second treatment—washed hydrothermally treated pulp (W-HTP)—the pulp was ground using the same conditions mentioned for the HTP samples. Subsequently, this material was washed with distilled water until quercetin was eliminated. After those treatments, the films were prepared by casting. For this, the aqueous suspension containing the onion pulp was poured into Petri dishes and dried at 40 °C for 6 h in a heat chamber (Quimis, Diadema, São Paulo, Brazil).

### 2.2. Co-60 γ-Radiation

The HTP and W-HTP films were irradiated with γ-rays in different doses—5, 10, 15, 25, 30, and 40 kGy—at the Gamma Radiation Laboratory (GAMALAB, Department of Nuclear Energy, Federal University of Pernambuco—UFPE). A Co-60 irradiator (Gammacell GC220, MDS Nordion, Ottawa, ON, Canada) was used, with a γ-energy of c.a. 1.25 MeV, under a rate of 1.44 kGy/h.

### 2.3. Film Characterizations

Field emission scanning electron microscopy (FE-SEM)—The HTP and W-HTP film morphology was analyzed by a FE-SEM microscope (JSM-7500F, JEOL, Tokyo, Japan) in secondary electron mode. 2 kV of acceleration voltage was applied, as well as 10 µA of emission current, with a working distance of c.a. 8 mm.

Differential scanning calorimetry (DSC)—The thermal transitions of the samples were investigated via DSC (DSC 1 STARe, Mettler Toledo Ind. e Com. Ltda, Barueri, SP, Brazil). The experimental conditions were static air atmosphere, sample mass around 6 mg in an open aluminum crucible, and heating rates of 10 °C min^−1^ from room temperature to 400 °C.

Thermogravimetry (TG) and derivative TG (DTG)—The thermal behavior of onion film samples was also studied via TG and derivative TG using a simultaneous thermal analysis module—SDT equipment (Q 600, TA Instruments, New Castle, DE, USA). The analysis was performed in an inert atmosphere with a rate of 100 mL min^−1^. About 5 mg of the films were weighed in an Al_2_O_3_ crucible (precision: 0.1 µg), and their thermal changes were evaluated from room temperature to 500 °C at 20 °C min^−1^.

Fourier transform infrared spectroscopy (FTIR)—The chemical composition was qualitatively investigated through FTIR, using an attenuated total reflection (ATR) accessory (Cary 630, Agilent Technologies, Santa Clara, CA, USA). The data were collected in transmittance mode for wavenumbers from 4000 to 800 cm^−1^ with a resolution of 4 cm^−1^. For each sample, 32 scans were performed.

X-ray diffraction (XRD)—The HTP and W-HTP film structures were analyzed via X-ray diffraction (XRD-6000, Shimadzu, Japan). The diffractograms were obtained for Bragg angles (2Ɵ) between 5 and 35° at a scan speed and a resolution of 4° min^−1^ and 0.02°, respectively. The diffractometer was operated using CuKα radiation at 40 kV and 30 mA. The curves were deconvoluted considering a Gaussian Function, aiming to distinguish the crystalline peaks from the amorphous halos and then determine the crystallinity index (CrI) according to Equation (1), where I_(200)_ and I_am_ represent the maximum intensity of the crystalline peak related to (200) plane and the minimum intensity between the peaks of (110) and (200) for 2Ɵ ≈ 18°, in this order [35].
(1)$$\mathrm{CrI}\;\left(\%\right)=\frac{I_{\left(200\right)}-I_{\mathrm{am}}}{I_{\mathrm{am}}}$$

Mechanical properties—The mechanical properties of rectangular films, measuring 60 mm × 20 mm at room temperature, were evaluated by a uniaxial tensile assay (DL-2000, Emic, Paraná, Brazil). The test configuration consisted of a 50 kgf load cell, 10 mm of initial distance, and a crosshead speed of 0.83 mm s^−1^. The ultimate tensile strength, elongation at break, and Young’s modulus were obtained from the Tesc v3.04 software.

Apparent contact angle measurements—The onion-based film’s wettability was evaluated using a goniometer (260 F4, Ramé-Hart, Succasunna, NJ, USA). The test was conducted in a static mode, using distilled water as the probe liquid. The data were calculated automatically by the DROPimage Advanced V2.7 software and were expressed as an average between nine different measurements, taken at intervals of 0.001 s per measurement after the water droplet deposition on the film surfaces.

Barrier properties—The analysis set for the study of the onion films’ barrier properties included the coefficient of permeability (α), water vapor transmission rate (WVTR) and water vapor permeability (WVP). For the WVRT experiments, 5 g of anhydrous CaCl^2^—previously dried at 200 °C for 1 h—were placed inside flasks (Ø = 30 mm; height = 75 mm) having perforated threads. The HTP and W-HTP films were cut into a circular shape with a diameter of 15 mm and placed on the open surface of the flasks. Furthermore, the flasks were inserted into a reactor containing 100 mL of NaCl aqueous solution and maintained at a temperature of 30 ± 2 °C, and the vials were weighed every hour for 8 h. The WVTR was determined, through Equation (2), in which α and A represent the mass of moisture permeated during this time (g h^−1^) and the area of the sample available for permeation (m^2^), respectively.
(2)$$\mathrm{WVTR}=\frac{\alpha }{A}$$

The WVP was calculated according to a modification of the gravimetric cup methodology described by McHugh et al. [36], which is based on the ASTM E96M-16 standard [37]. For this property, the thickness of the films (e) was considered in the calculus, which was completed by Equation (3). RH and Pv are the relative humidity (c.a. 75%), and the vapor pressure (31.824 mmHg), in this order.
(3)$$\mathrm{WVP}=\frac{100\;\cdot \;\mathrm{WVTR}\;\cdot \;e}{\mathrm{Pv}\;\cdot \;\mathrm{RH}}$$

Mutagenicity assays—The mutagenicity assays were conducted according to the pre-incubation methodology developed by Maron et al. [38], following the ISO 10993-12 for preparing the samples [39]. Firstly, the films were cut into pieces with an area of 6 cm^2^ and incubated with 1 mL of buffer solution at 37 °C for 72 h to obtain the eluate. Furthermore, 100 µL of this eluate was evaluated in the *Salmonella typhimurium* strain—TA98 and TA102—to eluates from films with different doses of radiation, 0.5 mL of 0.2 M phosphate buffer and 0.1 mL of the prepared *Salmonella typhimurium* culture were added. These mixtures were incubated at 37 °C for 20–30 min. After this period, 2 mL of top agar—supplemented with traces of L-histidine and D-biotin—were added to the mixture and poured onto a plate containing minimal agar. Finally, the plates were incubated at 37 °C for 48 h, and the number of revertant colonies in each place was manually determined. As positive controls, 4-nitro-o-phenylenediamine (10 µg/plate), for strain TA98, and mitomycin C, for strain TA102 (0.5 µg/plate), were used. The negative control had no treatment and consisted of the spontaneous reversal rate of each strain. The results were statistically analyzed using the GraphPad Prism 7 statistical program (Graph-Pad Software Inc., San Diego, CA, USA), by one-way analysis of variance (ANOVA), complemented by the Dunnett’s post hoc test. This test was performed in comparison with the control group (untreated control of the assay), and the mutagenicity index (MI) was calculated according to Equation (4), with the average number of revertants per plate with the test compound divided by the average number of revertants per plate with the negative control. A sample is considered mutagenic if the ANOVA variation is significant, with *p*  <  0.05, and the average number of revertants increase in the sample a minimum of two folds of that found in the untreated control (MI  >  2) [39].
(4)$$\mathrm{MI}=\frac{\mathrm{number}\;\mathrm{of}\;\mathrm{revertants}\;\mathrm{per}\;\mathrm{plate}\;\mathrm{with}\;\mathrm{the}\;\mathrm{test}\;\mathrm{compound}}{\mathrm{number}\;\mathrm{of}\;\mathrm{revertants}\;\mathrm{per}\;\mathrm{plate}\;\mathrm{with}\;\mathrm{the}\;\mathrm{negative}\;\mathrm{control}}$$

Cytotoxicity assays—Resazurin hydrochloride (Sigma-Aldrich, St. Louis, MI, USA) was used as a revealing substance, which has redox (oxide-reduction) potential and a colorimetric change and fluorescence indicator in response to cellular metabolism, according to the protocol of Page et al. [40]. The human keratinocyte cell line (HaCat) was kept as a monolayer in 25 or 75 cm^2^ culture flasks (Corning) at 37 °C in an atmosphere of 5% carbon dioxide (CO_2_) and 95% air, under saturated humidity, and grown in Dulbecco’s Modified Eagle’s Medium (DMEM medium) supplemented with 10% fetal bovine serum. Furthermore, HaCaT cells were seeded in 96-well microplates at a density of 1.0 × 104 cells/well, followed by incubation for 24 h for cell adhesion. For treatment, the eluates were previously prepared in culture medium (films with an area of 6 cm^2^ incubated with 1 mL of culture medium at 37 °C for 24 h [38]) and added to a 96-well plate with the cells grown in monolayers. Subsequently, serial dilution was performed along the plate, thus obtaining concentrations of 100, 50, 25, 12.5, 6.25, and 3.125% of eluate. Negative (no treatment) and positive controls (50% dimethylsulfoxide, DMSO) were also included. After the treatments, the cells were incubated for 24 h. Furthermore, the culture medium was removed, and 50 µL of 0.01% (*m*/*v*) resazurin hydrochloride were added and incubated for 4 h. The fluorescence was read in a spectrofluorimeter (Cary Eclipse, Agilent Technologies, Santa Clara, CA, USA) at excitation and emission lengths of 560 and 590 nm, respectively. The results obtained were expressed as a percentage of the negative control, which was considered to have 100% cell viability. Data were verified for normality by the K-S test (Kolmogorov-Smirnov test) and submitted to an analysis of variance (ANOVA) followed by the Dunnett comparison post-test, with the negative control as a reference. Statistical analysis and graphics were performed using the GraphPad Prism 9 program for Windows 10 (Intuitive Software for Science, San Diego, CA, USA).

## 3. Results

### 3.1. Film Morphology—Field Emission Scanning Electron Microscopy (FE-SEM)

The effect of pulp washing and the application of different doses of γ-radiation on the morphology of onion-based films were evaluated via FE-SEM (Figure 1). Firstly, for the non-irradiated materials, the unwashed samples present a denser, more homogeneous, uniform, and smoother surface compared with the W-HTP film. The latter exhibit several surface irregularities that result in increased roughness, but neither cracks nor voids exist. In addition, Dias et al. [3] previously analyzed the cross-section of these materials, and the results indicated a more cohesive fracture surface for the HTP films. Washing led to distinct morphologies in the upper and lower strata, in which irregularities and the presence of some cracks were visualized, respectively [3]. This change in morphology is attributed to the presence of soluble carbohydrates and acids in the chemical composition of HTP films, which act as plasticizers [3,41]. As already studied, these compounds mainly consist of glucose, galactose, arabinose, and fructose, besides formic, galacturonic, and acetic acids [3]. In addition, plasticizers increase the distance and relative mobility between macromolecules, providing greater workability and processability to biopolymers, among other characteristics and properties [41,42,43].

Regarding the effect of film radiation, two distinct behaviors were observed according to the applied dose. First, compared with non-irradiated samples, after the incidence of γ-radiation, the film surfaces became more homogeneous and less rough for doses up to 15 kGy, with a lower presence of sharp reliefs. On the other hand, doses equal to or higher than 25 kGy provided an increase in the surface roughness, which is more pronounced for the washed samples. For the W-HTP samples, the radiation caused the formation of some agglomerates, whose proportion increased according to the irradiated dose, becoming more evident above 25 kGy. In line with these results, clusters were also observed for the HTP films, but they were more dispersed along the surface, with some uniform and homogeneous regions remaining. Similar findings have been described in the literature for other biopolymers, such as chitosan [44], silk fibroin [45], and zein [46].

In general, ionizing radiation—such as γ-radiation—can penetrate materials and produce free radicals that can lead to the modification or breakage of the molecules that compose them [47,48]. Furthermore, these free radicals can react with the oxygen present in some chemical groups, causing film surface oxidation [49,50]. For example, for onion skin, Yang et al. [47] demonstrated a significant increase in the yield of quercetin and phenolic compounds after the incidence of 10 kGy of γ-radiation. However, smaller doses were insufficient to break physical and chemical bonds or to complex phenolic compounds with other compounds [47]. Thus, due to this oxidation, there is a change in several surface properties, such as roughness, chemical composition, and contact angle, which will be analyzed subsequently.

### 3.2. Fourier-Transform Infrared Spectroscopy (FTIR)

Qualitative information about the chemical composition of onion films was obtained via FTIR for wavenumbers between 4000–2000 cm^−1^ (Figure 2a,c) and 2000–800 cm^−1^ (Figure 2b,d). This analysis provided important changes in response to washing and γ-radiation dose. The main vibrations identified by this analysis refer to the constituent groups of cellulose, pectin, and xylans, in addition to mono-, di-, and polysaccharides [51]. These data have already been summarized in the literature [3].

In the spectra at 4000–2000 cm^−1^, all bands identified in the HTP films were shifted to lower wavenumbers after washing the pulp (Figure 2a). The most significant differences in response to washing are found at 3289–3269 cm^−1^ and 2922–2896 cm^−1^ for the W-HTP samples (Figure 2c), which originate from different types of ν(OH)—such as free OH, OH intra- and intermolecular H-bond, and OH-H hydrogen bonds next to cellulose ß-glycosidic bond—and ν(CH) mainly from carboxylate groups, respectively [34,52,53,54,55]. The presence of two adjacent bands, near 2365 cm^−1^, is associated with atmospheric CO_2_ and is practically absent for doses of 5, 15, and 40 kGy [52].

For smaller wavenumbers, more pronounced changes can be distinguished. Regarding the effect of washing on non-sterilized samples, the most apparent change is located at 1737–1733 cm^−1^ (Figure 2c). This band is seen only in W-HTP samples and is attributed to the vibration ν(C=O) of polygalacturonic acid present in pectin (Figure 2d) [56,57]. For such films, there is also the disappearance of the band at 924–917 cm^−1^, referring to the β(CCH), ν(CO)ring, and ν(CC)ring of some mono- and disaccharides that were removed by washing the pulp. Subsequent vibrations, relative to the ν(CC) and β(CCH) of certain mono- and disaccharides (867 cm^−1^), in addition to the ν(CC) of α-glycosidic linkage of trisaccharide (820 cm^−1^), exhibit a decreased intensity as a result of washing [51].

A very small influence on the chemical composition was noticed after the irradiation of the films, mainly related to the displacement of the bands. As mentioned earlier, these bands are related to some ν(OH) and symmetric ν(CH_2_) vibrations of lipids and fatty acids, in this order [53,54,55,56,57,58,59]. Thus, low doses of radiation may have caused them to break, although the same effect was not observed for doses higher than 25 kGy. The HTP film showed a shift in the band associated with δ(HOH) of absorbed water in pectin and cellulose—initially at 1636 cm^−1^—for decreasing values as the radiation dose increased, until reaching 1617 cm^−1^ for 40 kGy [60]. The γ-radiation leads to the formation of radical species that act as potent oxidizing agents. These radicals are mainly derived from hydroxyls [61]. Due to this, changes associated with this group are more evident in the FTIR spectra.

### 3.3. X-ray Diffraction (XRD)

According to XRD diffractograms (Figure 3; shown only for 5 and 40 kGy), both onion-based films are characterized as semi-crystalline and therefore have crystalline and amorphous regions. After deconvolution, an amorphous halo between 9.6 and 9.8°, referring to hemicellulose, is identified. For higher values of 2Ɵ, there are two crystalline peaks at 14.8–16.3° and 20.3–22.1°, which are attributed to the crystalline phase of cellulose, specifically to the crystallographic planes (110) and (200), respectively [62,63]. The first interval of 2Ɵ suggests the occurrence of overlap between triclinic and monoclinic unit cells in cellulose I, while the second refers to the distance between the hydrogen bonds in this unit [62]. For both films, it was not possible to differentiate the broad amorphous halos related to lignin and hemicellulose for 10 < 2Ɵ < 15° [63].

After irradiation of the samples, the crystalline peaks become more intense and narrower, but changes are not observed in the amorphous halo. To elucidate this difference, the CrI of the films was determined (Table 1). The results indicated that the W-HTP samples showed a higher crystallinity index than the HTP ones due to the increased rigidity of the cellulose fibers that constitute these films [3]. In W-HTP films there is an elevation in the degree of crystallinity as the dose of γ-radiation was increased—except for intermediate doses, such as 15 and 25 kGy reaching a maximum value of 54.4% at 40 kGy. A similar effect was observed for the HTP samples at higher doses, since the crystallinity index also decreased for 15 and 25 kGy, changing between 4.8 and 6.7% for these doses, in this order. In line with the results obtained by FTIR, irradiation can lead to the cleavage of polymer chains and chemical bonds. Even so, it is not possible to state that there was a significant difference in the crystallinity between the groups. However, these chains can later rearrange themselves with a greater degree of organization, resulting in an increase in crystalline regions and a decrease in amorphous ones [64]. It is more likely to occur with shorter chains, since they present fewer restrictions on mobility and rearrangement [65,66].

### 3.4. Differential Scanning Calorimetry (DSC)

The thermal transitions of onion-based films were evaluated by DSC, and the temperature variation in response to washing and γ-radiation dose will be emphasized. In the temperature range analyzed, it is possible to notice differences in terms of the influence of washing the films, especially in T_dehyd_. As can be seen in Figure 4a, HTP samples do not exhibit a dehydration peak in the temperature range between 100 and 200 °C, unlike W-HTP films (Figure 4b), in which the peak, referring to this reaction, is clearly visible. Furthermore, the enthalpy variation involved in this reaction tends to increase with the applied radiation dose, except for 40 kGy (Table 2). Dehydration in this temperature range can be attributed to the removal of water—the main component of onion bulbs—and is also associated with the presence of soluble carbohydrates in HTP samples [65]. Some of them are crystalline compounds—such as glucose and fructose—and, therefore, lose their crystalline structure during heating at lower temperatures, which is the major difference between both films [66].

In addition, the curves contain exothermic peaks at temperatures between 188–229 °C and 300–371 °C. The first event is associated with the degradation of the amorphous structure of hemicellulose, producing char residues [67]. The second peak, in turn, is mainly related to lignin, with a minor contribution from cellulose. Lignin contains aromatic rings with several ramifications that allow the establishment of chemical bonds susceptible to the oxidative effect promoted by γ-radiation [68]. For W-HTP films, only the second exothermic transition is visualized and becomes more evident with higher radiation intensities, starting at lower temperatures and involving greater enthalpy variation. Therefore, it is indicated that γ-radiation favors the degradation of these lignocellulosic components, a result also obtained in other studies [69,70]. In contrast, for the HTP samples, there are no great changes in the results since the enthalpy and peak temperature remain similar for all doses.

### 3.5. Thermogravimetry (TG/DTG)

In addition to DSC, the thermal behavior of the HTP and W-HTP films was also evaluated by thermogravimetry (Figure 5). The HTP samples showed three distinct steps of mass loss, located in the following temperature ranges: 120–165 °C, 120–250 °C, and 250–400 °C (Figure 5a, 5b). Mass losses at temperatures below 120 °C, as observed for the non-sterilized sample, can be attributed to the loss of water and other volatile compounds (Figure 5b1) [3,71]. The second step begins with the decomposition of the onion’s organic components that have lower thermal stability, mainly hemicellulose (Figure 5b2) [72,73]. The third step, in turn, is associated with the decomposition of carbohydrates and cellulose, a macromolecule that has greater thermal stability compared with hemicellulose (Figure 5b3) [71,72]. Above 400 °C, the relative mass progressively decreases, a behavior related to the thermal decomposition of lignin (Figure 5b4). It is highlighted that the compounds mentioned are the major components subject to decomposition in these temperature ranges, but other structural carbohydrates, such as pectin and non-structural carbohudrates (free fructose, free glucose, sucrose, and fructans), as well as flavonoids, also decompose at such intervals [3,73,74,75].

Regarding the effect of the γ-radiation application, no significant changes were observed in the intervals in which there was a higher mass loss. For the first step, the films tended to have two maximum mass losses, so the first one moved to lower temperatures as the γ-radiation dose was increased. The opposite behavior occurred for the second maximum value. As an example, HTP, treated under 5 kGy, showed maximums at 133 and 144 °C, values that shifted to 128 and 148 °C for 30 kGy. Furthermore, at this stage, all irradiated samples showed higher mass losses compared with the non-irradiated sample. However, in the other stages, no relevant differences were observed, since all samples exhibited mass losses similar to those obtained for non-sterile HTP, but with a slight shift in temperature of the maximum mass loss to progressively lower values as the dose of γ-radiation rose.

Similar results were obtained for the W-HTP films, which presented an additional step of mass loss for temperatures between 50 and 130 °C (Figure 5c,d). This step is related to the elimination of volatile compounds and residual moisture present in the samples, the latter derived from the washing procedure (Figure 5d1). The second step is found at 150–250 °C and is more pronounced for samples irradiated at higher doses, specifically 30 and 40 kGy. For these doses, at 200 °C, there was a loss of 16 and 12% in relation to the initial mass, respectively (Figure 5d2). Significant differences associated with the applied radiation dose are noted for the two subsequent steps, at temperatures of 250–400 °C (Figure 5d3) and 420–600 °C (Figure 5d4). In the first interval, the non-irradiated and irradiated samples, with doses of up to 15 kGy, exhibit a more pronounced mass loss compared with the others, a behavior contrary to the last interval, in which the loss of films treated with radiation greater than 15 kGy is significantly more intense, increasing according to the dose and moving to lower temperatures. Similar findings were observed by Nascimento et al. [76] and Ito et al. [77] for bacterial cellulose membranes and black rice flour, respectively, in which there was a small increase in the maximum decomposition temperature for higher γ-radiation doses, due to structural changes caused after irradiation.

In this way, it is inferred that the W-HTP films—even with an additional mass loss stage—present greater thermal stability compared with unwashed samples; that is, for the same temperature, the relative mass is higher than the HTP ones. However, this stability tends to be improved as the dose of γ-radiation increases for both materials, especially at high temperatures.

### 3.6. Mechanical Properties

The mechanical properties—specifically the tensile strength, Young’s modulus, and elongation at break—were evaluated as a function of the γ-radiation dose using the uniaxial tensile test (Figure 6). The tensile strength was significantly higher for W-HTP films compared with HTP films (Figure 6a). This behavior is attributed to the presence of a few soluble carbohydrates—especially low molecular weight sugars—which behave as plasticizers and, therefore, increase the mobility of macromolecules [3]. As a result, the polymer chains are more susceptible to alignment in the direction of the applied stress, so a small level of stress is able to lead to the rupture of the specimens [78,79]. The γ-radiation, regardless of the dose, resulted in an increase in this property for all samples in relation to non-irradiated materials, but this increase does not occur proportionally to the applied dose. Maximum tensile strength was observed at doses of 5 and 25 kGy for the HTP and W-HTP samples, respectively. The degree of crystallinity of the samples, dependent on the γ-radiation dose, also influences the mechanical properties of the films. In general, the tensile strength tends to increase with higher degrees of crystallinity, but this was not observed in the samples [80,81]. For W-HTP films, for example, the maximum tensile strength occurs at 5 kGy, a sample that does not have the highest degree of crystallinity.

As expected from the tensile strength results, significantly higher values of elongation at break were determined for the HTP samples (Figure 6b). This increase is also due to the plasticizing effect of some components of the films, which increase their flexibility, distensibility, and extensibility [3]. Furthermore, this property tended to increase according to the γ-radiation dose for both samples. However, for 40 kGy, the maximization of elongation was observed at break for the HTP films, with variations of 12 and 9% compared with the elongation determined at doses of 5 and 10 kGy, respectively.

Opposite to the elongation at break, the W-HTP films were stiffer than the unwashed samples (Figure 6c), supporting higher stresses but with less elongation in the elastic deformation regime. However, in accordance with the other properties, although the application of radiation has increased the Young’s modulus of the W-HTP samples, especially for 5 and 10 kGy, the same behavior is not seen for the HTP samples. For these, the stiffness was similar to that of the non-irradiated onion-based film, with a maximum change of only 3% observed for 40 kGy. Thus, significant differences in mechanical properties are more related to washing, which leads to the brittle behavior of the films, with less effect of the γ-radiation dose.

### 3.7. Wettability—Apparent Contact Angle Measurements

The wettability—that is, the hydrophilic or hydrophobic nature—of the HTP and W-HTP surfaces was evaluated in response to the different doses of γ-radiation applied during the sterilization process (Figure 7). According to the Young-Laplace model, all films presented hydrophilic surfaces since the apparent contact angle upon deposition of a drop of deionized water was less than 90° [82,83]. This result was observed regardless of the washing process and the intensity of the radiation. However, a significant increase in the contact angle is indicated by comparing the values determined between the HTP and W-HTP samples for the same γ-radiation dose. Considering a dose of 5 kGy, the contact angle changes from 29 ± 4 to 44 ± 5° after washing, an increase in approximately 52%. This difference is maximized for 40 kGy, the highest intensity of γ-radiation used in this study, with a variation of 90%.

The distinction between the static contact angle results can be attributed to the effect of washing on the chemical composition and morphology of onion films, both of which have characteristics that directly influence their surface properties. Due to the presence of soluble carbohydrates—often hydrophilic compounds—in the chemical structure of HTP films, they tend to interact with the water droplet and partially absorb it [25]. Furthermore, such soluble carbohydrates, especially sugars, behave as natural plasticizers for the films and, therefore, increase the mobility of the polymeric chains [84]. This behavior, among other factors, depends mainly on the presence of OH groups available for the intermolecular hydrogen bonds [84,85]. Thus, the permanence of soluble carbohydrates leads to a greater number of hydroxyl groups on the film surface to interact with the water droplet and, therefore, allow its diffusion from the surface towards the interior of the materials produced [3]. Consequently, there is an increase in the hydrophilicity of HTP samples, resulting in smaller contact angles compared with W-HTP.

When evaluating the influence of different doses of γ-radiation on the contact angle, no tendency of variation is observed according to the increase in the applied dose for both samples. For HTP films, increasing the dose from 5 to 15 kGy made the film surface slightly more hydrophobic, increasing the contact angle from 29 ± 4 to 35 ± 2°, respectively. This value is reduced again for doses above 25 kGy, remaining similar to that obtained for the dose of 5 kGy and practically unchanged as the dose is increased up to 40 kGy, corresponding to 32 ± 3°. Regardless of the dose, the γ-radiation incidence made the onion film surfaces more hydrophilic than the non-irradiated surfaces.

In general, the incidence of γ-radiation on polymeric materials—synthetic or natural—induces the oxidation of their surfaces, causing them to be enriched with chemical groups that contain oxygen, such as hydroxyls, carbonyls, and carboxyls [86,87]. As a result, there is an increase in hydrophilicity—or a reduction in the apparent contact angle—especially for high radiation doses, in this case, greater than 25 kGy. The absence of tendency for the contact angle was also obtained for the washed samples. The contact angle was greater only when the γ-radiation dose increased from 5 to 10 kGy and from 25 to 40 kGy, with elevations of 5 and 12% for these intervals. Comparing those, it is concluded that, although higher doses of radiation induce surface oxidation and change in the roughness of the films it was not sufficient to reduce the contact angle of the W-HTP samples to the level presented by HTP [82,88].

### 3.8. Barrier Properties

In addition to the morphological, chemical, structural, and thermal behavior changes, it is also important to evaluate the variation of the barrier properties in relation to the γ-radiation dose (Figure 8). It is noted that α and WVTR were higher for the washed onion films compared with the others, regardless of the intensity of γ-radiation (Figure 8a,b). Furthermore, these two barrier properties showed an elevation in their values compared with nonirradiated samples and similar changes with respect to the γ-radiation dose, but different for the HTP and W-HTP samples. For HTP films, with the exception of samples treated at 10 kGy and 30 kGy, a progressive increase in properties was observed for doses equal to or greater than 15 kGy. Despite this elevation, even at high doses of γ-radiation, the properties remain slightly greater than those of the untreated film. Alternatively, the opposite behavior was observed for W-HTP films. After an initial increase of 70% for both properties, the values of α and WVTR showed little difference up to 15 kGy. However, they decrease for 25 and 30 kGy and are elevated again for 40 kGy, achieving values of 0.0027 ± 0.0001 g h^−1^ and 28.42 ± 1.05 g h^−1^ m^−2^, respectively.

Regarding WVP, an opposite effect was observed compared with the other barrier properties (Figure 8c). First, the WVP values were higher for the HTP films and, in general, showed an increase after the application of 5 and 40 kGy relative to the non-irradiated reference sample. However, this increase was not proportional to the dose, since it was reduced to doses between 10 and 30 kGy, the lowest value observed for the second one, corresponding to 0.0654 ± 0.0003 g mm mmHg h^−1^ m^−2^. For W-HTP films, this property was reduced in response to the intensity of γ-radiation, especially for doses higher than 25 kGy. In short, the changes in the barrier properties were more pronounced for samples irradiated with doses greater than 15 kGy, with the W-HTP films being the most affected by the incidence of γ-radiation.

### 3.9. Mutagenicity Assays

The results of the evaluation of the mutagenic potential of the irradiated films in the genetically modified strains of *S. Typhimurium* TA98 and TA102 are presented in Table 3. The results were validated using untreated and positive controls.

The test system chosen in this study is an assay specifically designed to detect a wide range of chemical substances that can produce genetic damage that leads to gene mutations. The Salmonella strains are histidine dependent and carry different mutations in various genes in the histidine operon, which act as hot spots for mutagens that cause DNA damage via different mechanisms [31]. The strain TA98 detects frameshift mutations, and TA102 is normally used to detect mutagens that cause oxidative damage and base-pair substitution mutations (AT) [29].

This study shows that the HTP and W-HTP onion films are not mutagenic after the application of different doses of radiation. The treatments did not induce any statistically significant increase in the number of revertants when compared with the negative control, and the MI was lower than 2.0, demonstrating the absence of mutagenic activity under the conditions used in this study. In addition, none of the evaluated samples showed a toxic bactericidal effect, as a statistically significant reduction in the number of colonies was not observed compared with the spontaneous rate of bacterial reversion (negative control).

### 3.10. Cytotoxicity Assay

Cell viability results showed that, after the 24-h period, HTP films (Figure 9a) were more cytotoxic than W-HTP films (Figure 9). HTP films show a statistically significant reduction in the cell viability percentage of HaCat cells compared with the negative control (without any treatment) at 25, 50, and 100% concentrations, while changes in the percentage of viable cells after treatments with the W-HTP films were not statistically different from the negative control.

## 4. Conclusions

In this work, onion-based films (*Allium cepa* L.) were produced using two different hydrothermal treatments, aiming at evaluating the influence of γ-irradiation on their morphologies, properties, and performance. Irradiation was carried out in order to sterilize the films, with the knowledge of the γ-radiation effect dose essential for the prediction of their behavior prior to their application. The results indicated a more pronounced change in the properties of the films as a function of pulp washing than the applied radiation dose. High doses, such as 40 kGy, provided a reduction in the uniformity and homogeneity on the film surfaces, together with an increase in roughness and hydrophobicity. Such changes occurred due to the oxidative effect promoted by γ-radiation, which was also indicated by the cleavage of some chemical bonds, and changes in the crystallinity index and thermal and mechanical behaviors. Although both samples showed an absence of cytotoxicity, the HTP films exhibited a decrease in cell viability for eluate concentrations higher than 25%, different from the results obtained for W-HTP, in which no cytotoxicity was observed. In summary, our findings showed that the W-HTP properties were more significantly affected by the incidence of γ-radiation than the ones of HTP films, especially at doses higher than 15 kGy, and the performance of the onion-based films is highly dependent on the dose applied.

## Figures and Tables

**Figure 1 polymers-15-00914-f001:**
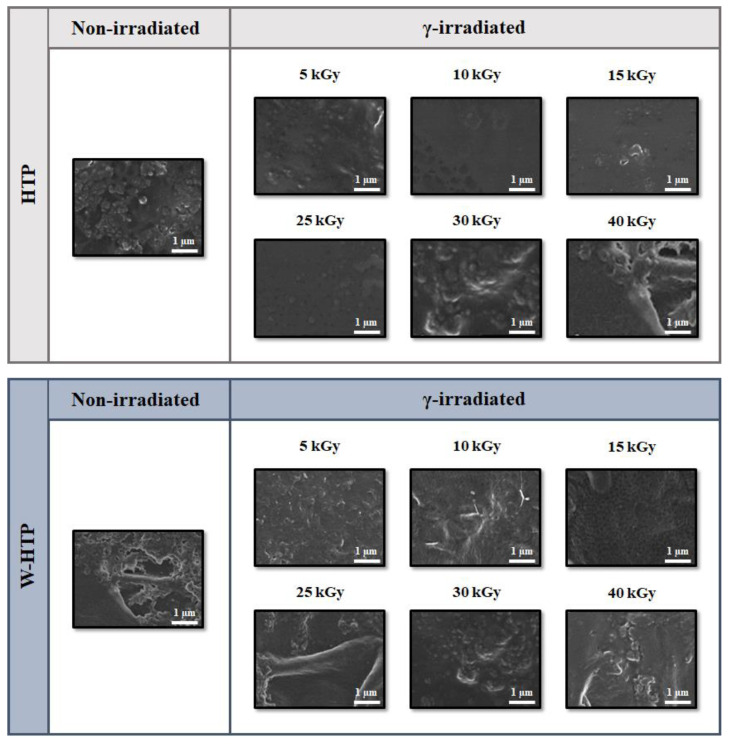
Micrographs obtained via FE-SEM of non-sterilized HTP and W-HTP onion-based films irradiated with doses between 5 and 40 kGy of γ-radiation (Magnification: 25,000×).

**Figure 2 polymers-15-00914-f002:**
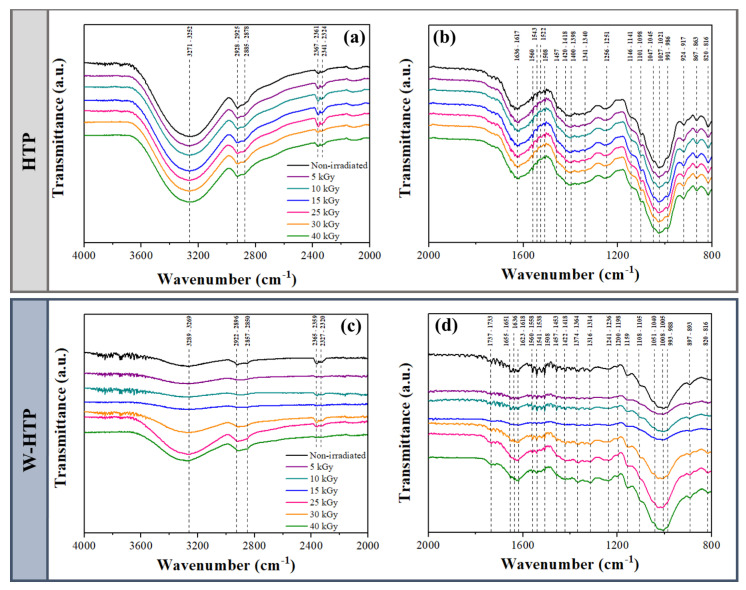
FTIR spectra of onion-based films for wavenumbers of 4000–2000 cm^−1^ (**a**,**c**) and 2000–800 cm^−1^ (**b**,**d**).

**Figure 3 polymers-15-00914-f003:**
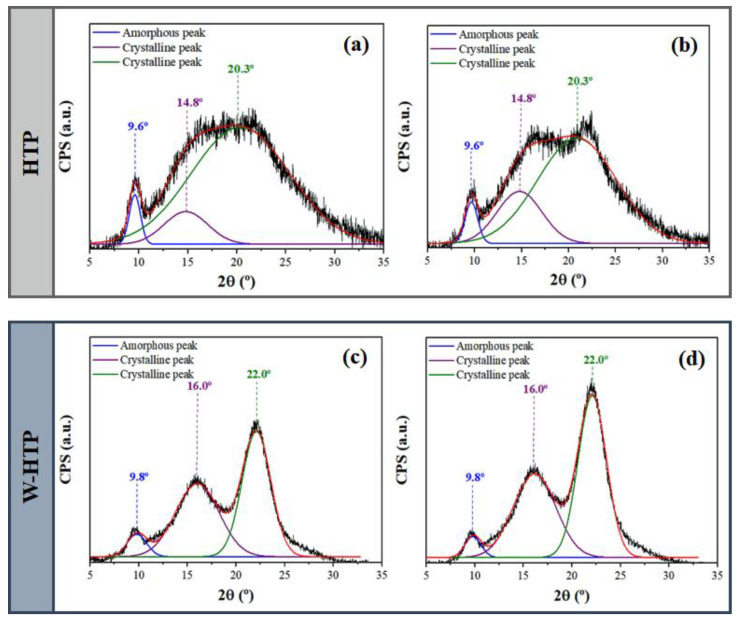
X-ray diffractograms (XRD) of HTP and W-HTP films sterilized with doses of 5 (**a**,**c**) and 40 kGy (**b**,**d**) of γ-radiation.

**Figure 4 polymers-15-00914-f004:**
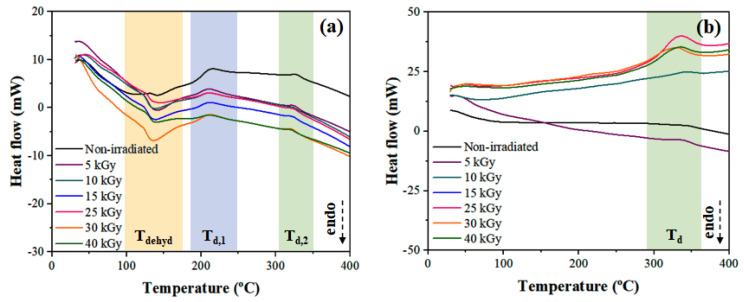
DSC curves showing the main thermal transitions of the onion-based films: (**a**) HTP; (**b**) W-HTP.

**Figure 5 polymers-15-00914-f005:**
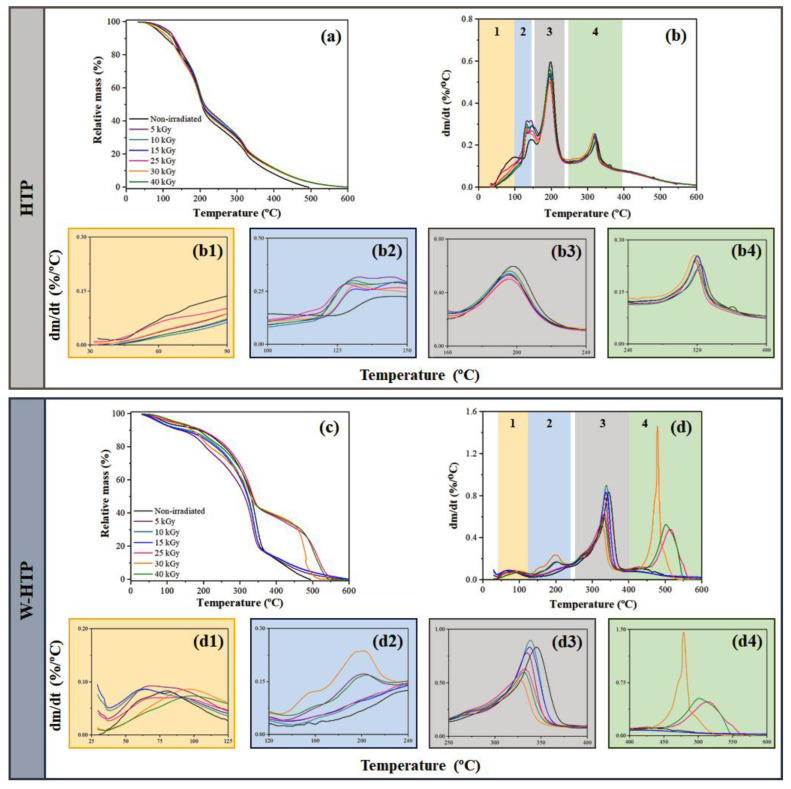
Thermogravimetric (TG) and derivative (DTG) curves of HTP (**a**,**b**) and W-HTP (**c**,**d**) onion films sterilized under different doses of γ-radiation. The stages of weight loss related to the major components of HTP (**b1**–**b4**) and W-HTP films (**d1**–**d4**) are also depicted.

**Figure 6 polymers-15-00914-f006:**
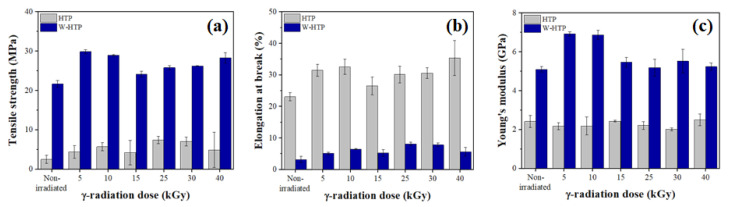
Mechanical properties of HTP and W-HTP films: (**a**) tensile strength; (**b**) elongation at break; (**c**) Young’s modulus.

**Figure 7 polymers-15-00914-f007:**
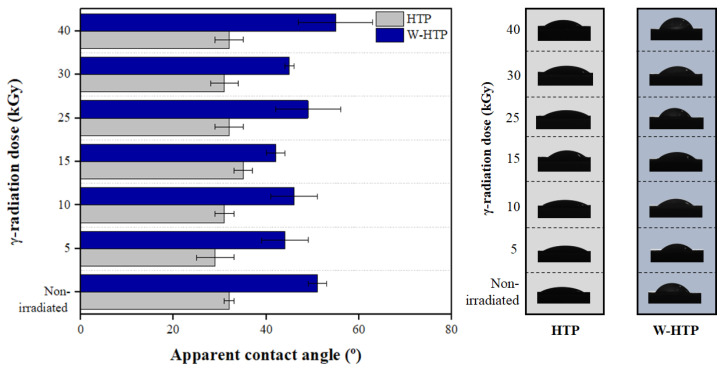
Apparent contact angle of the HTP and W-HTP γ-irradiated onion film surfaces under different doses and images of the water droplets 0.001 s after its deposition.

**Figure 8 polymers-15-00914-f008:**
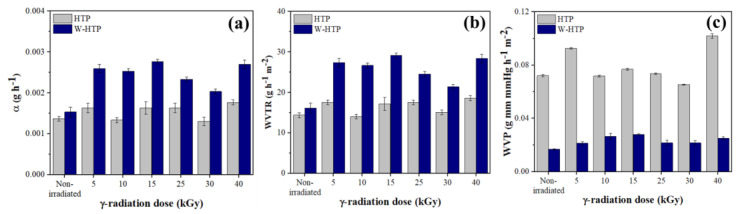
Barrier properties of HTP and W-HTP onion films under different radiation doses: (**a**) coefficient of permeability—α; (**b**) water vapor transmission rate—WVTR; (**c**) water vapor permeation—WVP.

**Figure 9 polymers-15-00914-f009:**
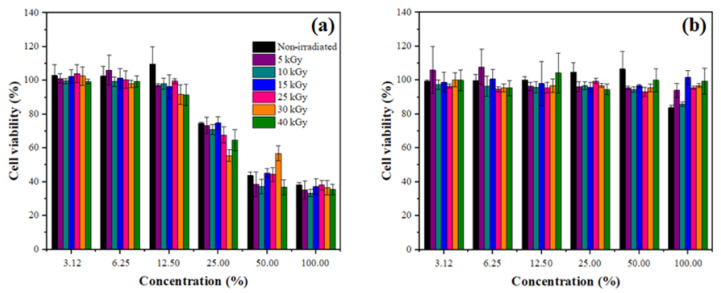
Analysis of cells viability in HaCat exposed to different concentrations of HTP onion films for 24 h from three independent experiments for the HTP (**a**) and W-HTP (**b**) samples.

**Table 1 polymers-15-00914-t001:** Intensity of the crystalline peaks—named as I_(200)_ and Iam—and crystallinity index (CrI) of the HTP and W-HTP films irradiated with doses between 5 and 40 kGy of γ-radiation.

Sample	γ-Radiation Dose (kGy)	I_(200)_ (2Ɵ ≈ 22°)	I_am_ (2Ɵ ≈ 18°)	CrI (%)
HTP	5	966	914	5.38
	10	1068	988	7.40
	15	910	866	4.80
	25	926	890	4.00
	30	1014	946	6.70
	40	1074	966	10.0
W-HTP	5	2138	1084	49.0
	10	1696	826	51.2
	15	1656	972	41.3
	25	1636	964	41.7
	30	1760	902	48.7
	40	2712	1236	54.4

**Table 2 polymers-15-00914-t002:** Values of dehydration (T_dehyd_), and crystallization (T_c_) temperatures—including the initial (T_onset_) and end (T_endset_) temperatures of these transitions—for the HTP and W-HTP films irradiated with doses between 5 and 40 kGy of γ-radiation.

Sample	γ-Radiation Dose (kGy)	T_dehyd_			T_d_		
		T_dehyd,onset_ *	T_dehyd,endset_ *	ΔH_dehyd_ **	T_d,onset_ *	T_d,endset_ *	ΔH_d_ **
HTP	Non-irradiated	132	154	8	229/342	188/318	−53/−9
	5	121	173	124	228/336	195/310	−18/−9
	10	121	169	97	225/338	192/317	−17/−5
	15	122	166	79	224/333	197/312	−16/−7
	25	119	159	78	225/335	195/315	−13/−7
	30	120	179	85	222/334	193/313	−24/−10
	40	118	171	42	227/332	193/312	−12/−6
W-HTP	Non-irradiated	-	-	-	357	326	−5
	5	-	-	-	354	320	−14
	10	-	-	-	368	324	−15
	15	-	-	-	369	300	−190
	25	-	-	-	371	317	−230
	30	-	-	-	356	302	−171
	40	-	-	-	359	316	−158

Units: * T_dehyd_ and T_d_ (T_onset_, T_endset_) in °C; ** ΔH_dehyd_ and ΔH_d_ in J/g.

**Table 3 polymers-15-00914-t003:** Mutagenic activity expressed as the mean (M) and standard deviation (SD) of the number of revertants/plate and mutagenicity index (MI) in *Salmonella Typhimurium* strains TA98 and TA102 after treatment with eluates of HTP and W-WTP onion films with γ-radiation.

		Revertants Number (M ± SD)/Plate and MI		
			TA98	TA102
		C−	23 ± 2	245 ± 49
		C+	862 ± 64 ^a,^*	1259 ± 102 ^b,^*
		γ-radiation dose (kGy)		
HTP (100 µL/plate)		5	23 ± 7 (1.0)	244 ± 23 (1.00)
		10	22 ± 0 (0.96)	249 ± 30 (1.02)
		15	22 ± 3 (0.96)	230 ± 14 (0.94)
		25	25 ± 4 (1.09)	265± 44 (1.08)
		30	31 ± 1 (1.35)	284 ± 12 (1.16)
		40	25 ± 4 (1.09)	260 ± 28 (1.06)
W-HTP (100 µL/plate)		5	30 ± 1 (1.28)	266 ± 23 (1.08)
		10	25 ± 1 (1.09)	236 ± 18 (0.96)
		15	24 ± 6 (1.04)	252 ± 40 (1.03)
		25	27 ± 4 (1.17)	257 ± 26 (1.05)
		30	26 ± 6 (1.13)	234 ± 37 (0.96)
		40	24 ± 6 (1.04)	230 ± 14 (0.94)

* *p* <0.05 (ANOVA); M ± SD = mean and standard deviation; MI = mutagenicity index (values between parentheses); Negative control (C−): no treatment (spontaneous rate of bacterial reversion); Positive controls (C+): ^a^ 4-nitro-o-phenylenediamine (10.0 μg/plate) and ^b^ mitomycin C (0.5 μg/plate).

## Data Availability

Not applicable.

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
