# Peer review of "Self-Supported Biopolymeric Films Based on Onion Bulb (Allium cepa L.): Gamma-Radiation Effects in Sterilizing Doses"

_polymers, 2023, doi:10.3390/polym15040914_

Round 1
Reviewer 1 Report
The work has novelty and it seems interesting.
The major differences obtained was in the washing process. It seems to be determinant in the final properties. The variations in the dose of radiation are not significant in almost all the measurements done. Because of that, the authors mustexplain the washing process in details and the effect in the final properties.
Moreover, the work has a lack of scientific rigor.
There are missing information and mistakes . For examples,
In the methods part:
Film characterization: Opacity? Macroscopic aspect? Transparency? Homogeneus film? Colored film? Thickness of the film?
In Mechanical properties: Samples sizes? Environments conditions? Sample shapes?
In results,
FTIR measurements: It is not clear for me…you wrote about the INTENSITY of the peaks, how do you obtain the FTIR spectra? Using ATR (as you put in the methods part?), So, you can not say anything about the intensity of the peaks-
DRX assays
You wrote: ¨ There is an elevation in the degree of 350 crystallinity as the dose of γ-radiation was increased – except for intermediate doses like 351 15, 25, and 30 kGy –, reaching a maximum value of 54.4% at 40 kGy. A similar effect was 352 observed for the HTP samples for higher doses, since the crystallinity index also decreased 353 for 15 to 30 kGy, changing between 4.8 and 6.7% for these doses, in this order. In line with 354 the results obtained by FTIR, irradiation can lead to the cleavage of polymer chains and 355 chemical bonds.¨
NO TENDENCY ARE FOUND! The DRX spectra are noisy and you can not conclude anything.
DSC curves,
It is impossible to see a Tg in these curves, you must do the DSC from -50°C or less. Or Maybe, you can do a reomethry test to show the Tg.
If you say the Tg start al 31 °C in some cases, the mechanical properties were measure in what temperature?
Reviewer 2 Report
1. In the "Introduction" section, the characterizations of onion-based film and γ radiation should be reviewed in more detail.
2. In this paper, the authors claim to have prepared a sterilize film, but have not analyzed the bactericidal properties of the film.
3. The application of the film needs further analysis and discussion.
Reviewer 3 Report
It is an interesting, well-documented and conducted study whose objective is to test the possibility of using onion-based bioplastics in the medical and food fields. For these applications, an important aspect is the influence of sterilization on the main properties of bioplastics: thermal, mechanical, the wettability, barrier properties, mutagenicity and cytotoxicity assays. The authors opted for radiation sterilization. Before publication, the authors should make a series of additions:
1. Them to specify in the introduction what are the other sterilization options that can be applied to bioplastics and to argue in more detail why they opted for radiation sterilization.
2. In section 3.5 it is observed that in the case of radiation doses higher than 25 kGy, the degradation mechanism changes. The authors only stated that other researchers have observed similar behaviors but did not explain why.
3. I suggest the authors to apply for comparison a test of the materials under constant temperature conditions (DOI: 10.1177/0954008312444294), respectively 120°C for different times.
Round 2
Reviewer 2 Report
1. In line 102, it is only mentioned that 25kGy is only the sterilized dose of health products, not for everything. Did you do any experiments to evaluate the antibacterial activity of your film?
2. Please analyze and discuss the application of your film according to its characteristics.
